# Planar polarity in primate cone photoreceptors: a potential role in Stiles Crawford effect phototropism

Anna Verschueren [1,2], Leyna Boucherit[1,2], Ulisse Ferrari [1], Stéphane Fouquet[1], Céline Nouvel-Jaillard[1,2], Michel Paques[1,2], Serge Picaud [1,5✉] & José-Alain Sahel [1,3,4,5✉]

Human cone phototropism is a key mechanism underlying the Stiles-Crawford effect, a psychophysiological phenomenon according to which photoreceptor outer/inner segments are aligned along with the direction of incoming light. However, such photomechanical movements of photoreceptors remain elusive in mammals. We first show here that primate cone photoreceptors have a planar polarity organized radially around the optical center of the eye. This planar polarity, based on the structure of the cilium and calyceal processes, is highly reminiscent of the planar polarity of the hair cells and their kinocilium and stereocilia. Secondly, we observe under super-high resolution expansion microscopy the cytoskeleton and Usher proteins architecture in the photoreceptors, which appears to establish a mechanical continuity between the outer and inner segments. Taken together, these results suggest a comprehensive cellular mechanism consistent with an active phototropism of cones toward the optical center of the eye, and thus with the Stiles-Crawford effect.

[1] Sorbonne Université, INSERM, CNRS, Institut de la Vision, 17 rue Moreau, F-75012 Paris, France. [2] CHNO des Quinze-Vingts, INSERM-DGOS CIC 1423, 28 Rue de Charenton, 75012 Paris, France. [3] Department of Ophthalmology, The University of Pittsburgh School of Medicine, 3550 Terrace Street, Pittsburgh, PA 15213, USA. [4] Fondation Ophtalmologique Rothschild, 29 rue Manin, F-75019 Paris, France. [5]These authors jointly supervised this work: Serge Picaud, José-Alain Sahel. ✉email: serge.picaud@inserm.fr; sahelja@upmc.edu

In 1937, Stiles and Crawford discovered that a light beam passing through the center of the pupil was perceived as more intense than an identical beam passing through the periphery of the pupil, corresponding to a lower light efficiency for incident light oblique to the retina[1]. At the cellular level, this directional sensitivity of the retina results from two properties of the photoreceptors: (i) the photoreceptors and especially their outer/inner segments act as a waveguide, (ii) these photoreceptor compartments are aligned toward the center of the pupil. Photoreceptors are indeed organized perpendicularly to the retinal surface at the macula, but they become oblique and lose this perpendicularity as they move toward the periphery, the degree of tilt increasing with eccentricity[2–4]. This graduated tilt is thought to be due to phototropism, rather than development, because the optimal direction of the Stiles-Crawford effect can change over a few days following eye patching, artificial displacement of the pupil, and cataract removal[5–8]. The waveguide properties of photoreceptors have been accurately modeled, and are currently considered to be the result of different cellular organization, for example mitochondrial presence and alignment in the inner segment[9]. In regard, little is known about the phototropism of photoreceptors, particularly in primates. The Stiles-Crawford effect showing promises in the functional assessment of retinal diseases resulting in photoreceptor disorganization[10], an understanding of this phototropism is becoming increasingly necessary.

A proof of phototropism existence was provided by the detection of transient photoreceptor phototropism (TRP) in the photoreceptor outer segment (OS), which bends transiently toward an incoming oblique light beam in both mice and *Xenopus*, by compressing the interdisc space on the most exposed side[11–15]. This observation gives a first hint as to how photoreceptors determine the direction of the incoming light, but it does not explain how the OS and inner segments (IS) align themselves stably in this new direction. Furthermore, the rapid onset of this phenomenon (~10 ms for frog and ~5 ms for mouse), its transient nature, and its predominance in rod OS are not fully consistent with the Stiles-Crawford effect, which was reported principally in photopic conditions dependent on cone activity[16,17]. This initial outer segment phototropism defining the direction of the incoming light suggests that a second mechanosensitive mechanism subsequently stabilizes the orientation within the inner segment.

Photoreceptors are known to express the Usher interactome, which is responsible for mechanotransduction during hearing, in the hair cells of the cochlea (for a recent review see Géléoc et al.[18]). In human patients, Usher gene mutations induce blindness as well as deafness. The deafness is caused by disruption of the hair cells mechanotransduction, but the functional role of Usher proteins in vision remains unclear. Hair cells express the Usher proteins in their stereocilia. Their function is to transduce the acoustic displacement of the stereocilia into the opening of an ion channel, ultimately leading to cell depolarization. In primate photoreceptors, Usher proteins are localized in calyceal processes, a structure extending around the OS from the OS/IS junction, in both the rod and cone photoreceptors and of unknown function[19]. These calyceal processes form a ring, including the photoreceptor cilium, and are homologous to the hair cell stereocilia. There has been some debate about the presence or absence of calyceal processes in mouse photoreceptors[19] and their absence has even been proposed as an explanation for the lack of a visual phenotype in mouse models of Usher syndrome[20]. Hair cells form four rows, with the V-shaped hair stereocilia all oriented in the same direction, highlighting a planar polarity of the epithelium crucial for its normal function, which is completely disrupted in hearing pathologies, and especially in mouse models of Usher syndrome[21]. In contrast, the photoreceptor mosaic planar polarity has been investigated only for the cone photoreceptor in the fish retina[22], where the cilium of the ultraviolet cones lies on the side of the optic nerve.

We investigated the macroscopic distribution of Usher proteins and cytoskeletal proteins on the retina, both using classical immunohistochemistry and to a very high 3D resolution, using expansion microscopy (ExM)[23], with a protocol adapted for the retina. Our data provide a novel comprehensive hypothesis explaining the phototropism underlying the Stiles-Crawford effect.

## Results

**Expansion microscopy of photoreceptors.** Usher proteins are expressed in the calyceal processes of photoreceptors in non-human primates[19]. We first investigated the structural organization of these proteins by localizing the USHER 1 (USH1) protein, protocadherin 15 (Pcd15), relative to the actin cytoskeleton. We used ExM to obtain super-high resolution. ExM principle is to physically magnify the observed tissue, to enhance final image resolution. This magnification is based on the synthesis of a swellable polyelectrolyte hydrogel throughout the specimen. The observed tissue is first immunostained, then infused with the chemicals required for acrylamide polymerization. After polymerization, an enzymatic digestion is used for mechanical homogenization of the tissue, and the specimen is then expanded by a factor of up to 4.5 in all directions, in an isotropic manner.

We located actin filaments by immunolabeling of an associated bundle protein, espin 1, because the phalloidin stain commonly used to visualize actin is not compatible with ExM. In classic immunochemistry (referred to as "noExM" below), Pcd15 immunolabeling formed a continuous C-shaped arc around the cone OS, as previously reported in non-human primate photoreceptors[19]. This Pcd15-positive C-shaped arc was associated with espin 1-immunopositive puncta (Fig. 1a, c–e). On 3D reconstructions, the Pcd15 arc extended in an open tube around the cone OS, whereas the espin 1-positive puncta formed lines extending well beyond the Pcd15 labeling into the IS (Fig. 1f, g). This distribution of espin 1 was consistent with the reported localization of actin filaments within the calyceal processes, with their roots anchored deep in the IS[24,25].

ExM techniques enlarged the primate retina tissue by a mean calculated expansion factor of $4.5 \pm 0.1$ SEM ($n = 8$ retinal samples), consistent with the findings of previous studies on the brain, pancreas, lung, and spleen[26]. Following ExM, the general staining pattern was preserved on sections (Fig. 1a versus b, 1c–e versus h–j) and on 3D reconstructions (Fig. 1f, g to k, l), consistent with an isotropic 3D expansion of the tissues. On closer examination of the immunolabeled structures, the ExM Pcd15 immunolabeling no longer appeared to form a continuous C-shape, but was instead distributed in clear puncta, juxtaposed in pairs to the inner side of an espin 1-immunopositive punctum (Fig. 1h). In 3D reconstruction, Pcd15-immunostaining did not form an open tube (Fig. 1f, g), but instead formed long filaments beneath the epsin-1 immunolabeling at the core of calyceal processes (Fig. 1k, l). Consistent with the reported function of Pcd15 in the tip-links of hair-cell stereocilia[27], our observations suggest that Pcd15 forms two rows of links between the membranes of calyceal processes and those of the cone OS microvilli. The espin 1-positive calyceal processes thus formed a so-called "C-shaped palisade", fixed by these protein links around the base of the OS. These 3D reconstructions also provided evidence to suggest that Pcd15 is not exclusively associated with calyceal processes, but also frequently extends to the base of the cone OS within the C-shaped structure of calyceal processes, in a plane slightly oblique to the cross-section of the photoreceptor

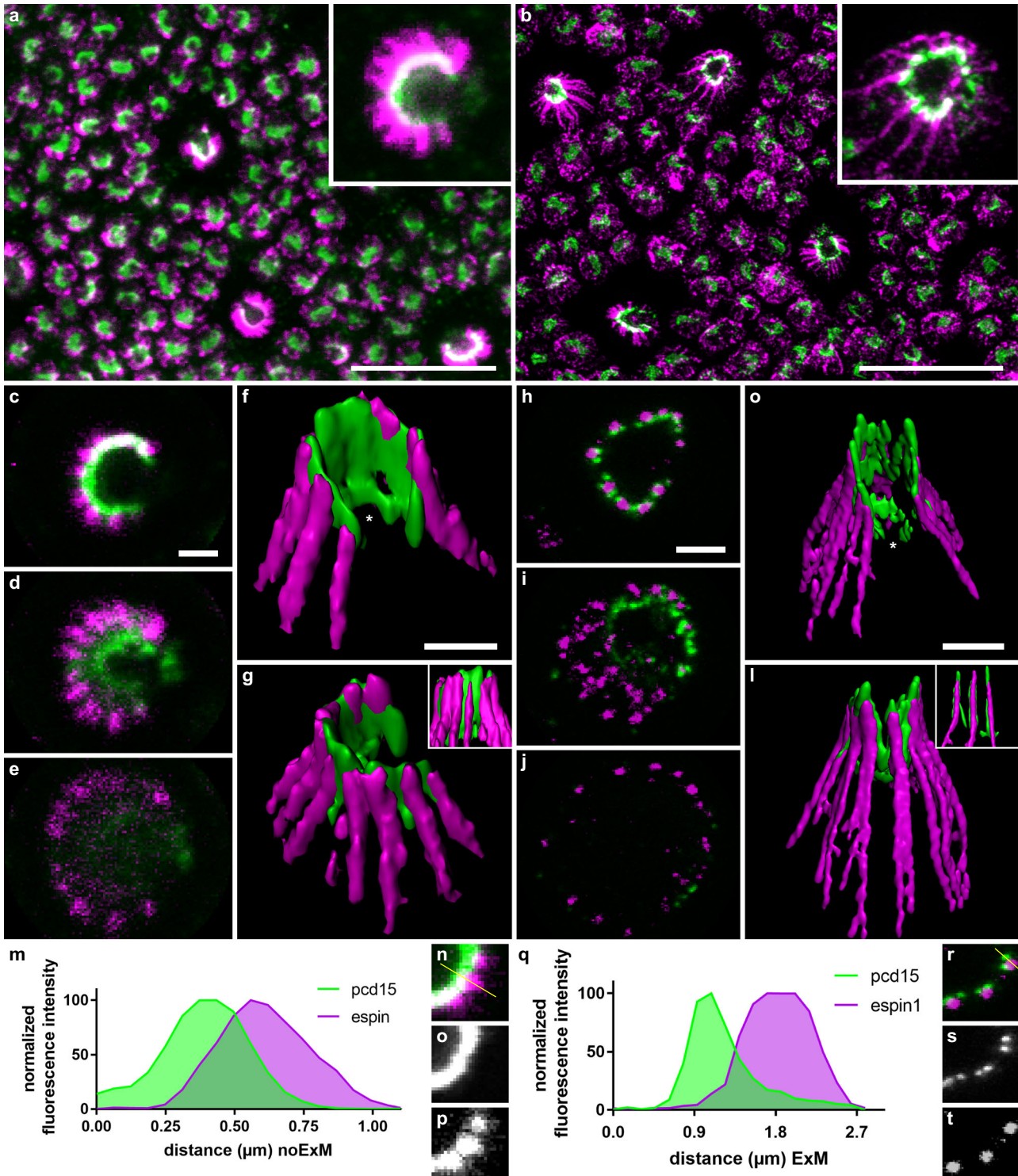

**Fig. 1 High magnification of cone calyceal processes by expansion microscopy.** Photoreceptor mosaics were examined on flat mounts of the retina by classic confocal microscopy (**a**, **c**–**g**, **m**–**p**) and by expansion microscopy (**b**, **h**–**l**, **q**–**t**). Photoreceptors were immunolabeled for espin 1 (magenta) and prodocadherin 15 (pcd15, green). **a**–**l** Immunolabeling of cone photoreceptors visualized in the photoreceptor mosaic as *z*-projections (**a**, **b**), on confocal optical sections from the outer to the inner segment (**c**–**e**, **h**–**j**) or on 3D reconstructions from different viewpoints (**f**–**g**, **k**–**l**). **m**–**t** Comparison of fluorescence profiles along a single calyceal process, showing the greater resolution of expansion microscopy (**r**–**t**) than of classic confocal microscopy (**n**–**p**) for espin 1 and Pcd15 immunostaining (numerical data in supplementary data 1). The profile section (yellow line) is shown on the isolated calyceal processes (**n**, **r**). Scale bars represent 10 µm in (**a**) and (**b**), and 1 µm in (**c**) to (**l**) (after division by the 4.5 expansion factor in **b**, **h**–**l**).

(see the white stars in Fig. 1f, k). This pattern was frequent, but was not systematically observed. Pcd15 thus probably also establishes links at the junction between the membranes of the OS and IS. For the sake of clarity, the scale bars in Fig. 1b, h–l, q–t represent the post-expansion size, whereas those in the subsequent figures are divided by the expansion coefficient of 4.5.

**The OS/IS junction in primate cone photoreceptors**. We defined the localization of Pcd15 at the cone OS/IS junction more precisely, by immunolabeling the retina with antibodies against the cone arrestin and the red/green (R/G) cone opsin (Fig. 2b–f). The cone arrestin antibody was used to determine the anatomy of whole cones, which was found to be well preserved after ExM (Fig. 2b), based on comparison with images obtained by scanning electron microscopy (Fig. 2a). On optical sections and reconstructed coronal sections (Fig. 2c, d), cone arrestin was located on the inner side of the Pcd15-positive cells beneath the calyceal processes. By contrast, the R/G cone opsin antibody labeled a tube at the opening of the Pcd15-positive C-shaped calyceal palisade (Fig. 2f). Surprisingly, at its base, the Pcd15 immunolabeling delineated a faint opsin-positive shape enclosed within a larger ring defined by the espin 1-positive puncta (Fig. 2f, yellow arrowhead). This inner structure at the OS/IS junction was highly reminiscent of the periciliary membrane at which the nascent cone OS microvilli form. For confirmation of this interaction between nascent OS microvilli and the Pcd15 protein, we immunolabeled the retinal tissue for Pcd15, R/G opsin and prominin 1, which is specifically expressed at the rim of nascent microvilli[28,29]. Figure 2j shows a classical confocal image, on which immunolabeling for prominin 1 and Pcd15 colocalized at the base of the cone outer segment, on the C shape opposite the opsin tube. The distribution profiles of the two proteins suggest that Pcd15 is located in a position external to prominin 1 (Fig. 2l). These observations are consistent with the conclusion that the Pcd15 protein links the membranes of nascent cone OS microvilli with both the membranes of calyceal processes and the periciliary membrane.

We visualized the position of the cilium, by labeling for the acetylated α-tubulin found in stable polymerized tubulin, and for centrin 3, a component of basal bodies[30]. The tubulin-positive structure formed a tube similar to that visualized by opsin staining, at the C-shaped palisade opening. Thanks to the higher resolution of ExM, we were able, with this technique, to distinguish the nine different tubulin doublets of the cone cilium (Fig. 2g). Centrin3-positive basal bodies were located at the base of the Pcd15 immunolabeling and at the C-shaped opening.

To investigate type 2 Usher protein, we immunolabeled VLGR1 (very large G protein-coupled receptor 1), which was reported to be located between the periciliary membrane and the cilium membrane in mice photoreceptors[31]. Here, in primate cone photoreceptors, we detected VLGR1 at the calyceal C-shaped opening in a ring of six to seven lobules (Fig. 2i). Again, the higher resolution of ExM elucidated this lobular pattern, which was previously observed by electron microscopy in mouse photoreceptors[31]. The lobules were located just below the Pcd15 labeling, establishing a link attaching the IS periciliary membrane to the OS cilium membrane (Fig. 2i). All these findings confirm that the intensely stained cone opsin/arrestin-positive tube delineates the OS structures located around the cilium. These observations suggest that the link between the OS and the IS is first formed by USHER2 proteins at its very base of the cilium (surrounding the basal body), and then above by USHER1 proteins (Pcd15), from the periciliary membrane to the tips of the calyceal processes, where the nascent cone OS microvilli appear (see Fig. 7a for a 3D schematic representation of

all proteins' expression). This tight bonding at the OS/IS junction, mediated by Usher proteins and strengthened by the sealed calyceal processes in a C-shaped palisade opposite the cone cilium, is probably necessary to the OS/IS optic waveguide property, avoiding a 'bend optic fiber' effect and preserving the IS/OS alignment.

**A motile system in the inner segment of primate cones**. Surprisingly, when we immunolabeled cone photoreceptors with the acetylated α-tubulin antibody to visualize the cilium, a genuine cortex of tubulin filaments was observed in the IS, intermingling with the cortex of actin filaments. Indeed, the distribution of espin 1 labeling underlying the actin filaments formed a cortex of distinct filaments anchored in the apical half of the IS (Fig. 3a, b also visible in **j** and **k**). Similarly, the tubulin filaments extended throughout the IS, elegantly delineating the morphology of the cone IS (Fig. 3g–l) and appear anchored in the basal half of this compartment. However, the density of acetylated α-tubulin filaments decreased below the calyceal palisade, in which the density of actin filaments was high. At intermediate positions in the IS, these two types of filaments appeared to be in close contact, as their distributions partially overlapped even after ExM of the tissue (Fig. 3k–m). This overlapping distribution is consistent with their known contribution to cell motility, in which actin filaments are intertwined with microtubules to generate relative movement driven by molecular motors, such as dynein or kinesin. These intermingled cortices of the two filaments suggest a possible major role in IS phototropism, as reported for retinomotor movement in *Xenopus* photoreceptors[32].

We then investigated the distribution of another intermediate filament previously described in mouse photoreceptors, rootletin (CROCC in Fig. 3g–j). We found that rootletin filaments ran up to the base of the IS, forming a column in direct continuity with the acetylated α-tubulin doublets within the cilium (Fig. 3l). These rootletin columns contained three to five filaments at their apical end, but only one at the base (Fig. 3h). Rootletin filaments are known to be stable in mature cells, and to not undergoing active polymerization/depolymerization cycles[33,34], like the tubulin filament doublets in the cilium. Their coherent alignment therefore generates a stable mast on the edge of the OS/IS. In addition, this mast is anchored on another perpendicular rootletin filament in the plane of the outer limiting membrane (Fig. 3g, h).

This tubulin/rootletin mast, the dense IS cortices of cytoskeleton filaments, and the strong Usher protein OS/IS junctions suggested that the OS/IS junction formed a very rigid and stable structure. This hypothesis was strengthened by a common observation in whole-mounted retina. When mounting retina in between coverslips for microscopic observations, the protocol unavoidably induced an uncontrolled shear-stress on the fixed sample. We observed that the shape of all the inner segments was well preserved, with strict linearity and all segments parallel to each other, despite the manipulation of the tissue and the pressure exerted by the coverslip (Fig. 4). When the OS was misaligned with respect to the IS, the calyceal palisade was maintained in correct alignment to the IS (Fig. 4b). Misalignment mostly occurred above the calyceal processes. Although this observation on fixed tissues cannot represent the in situ live conditions, it suggests that a mechanical link was present at the OS/IS junction and that the IS is also mechanically very stable (Fig. 4b–d).

**Planar polarity in the primate cone mosaic**. The C-shaped calyceal palisade strongly resembles the V-shaped structure of hair cell stereocilia. We therefore investigated whether the

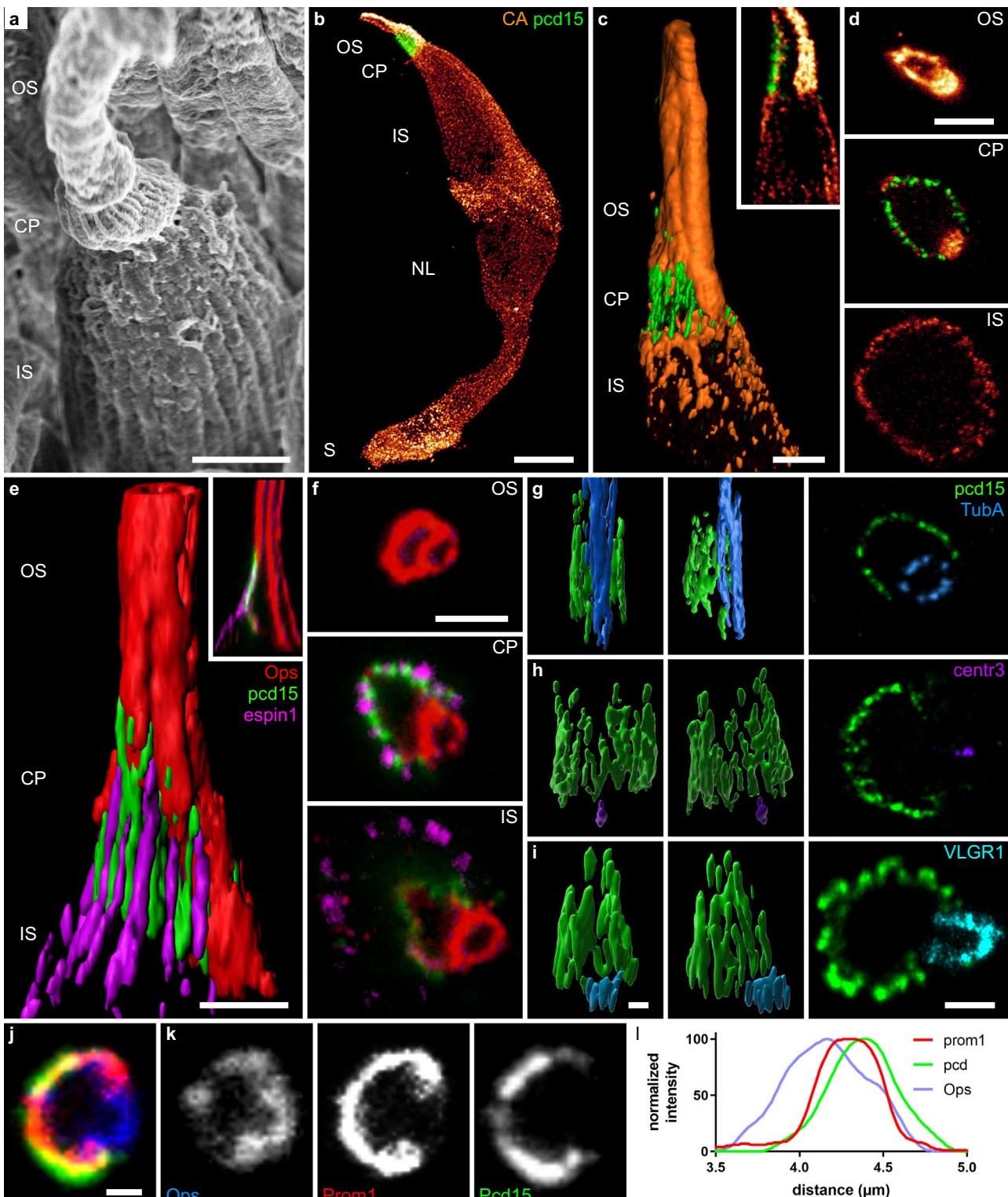

**Fig. 2 Calyceal C-shaped palisade around the cone cilium. a** Scanning electron microscopy of a primate cone showing the outer and inner segment as well as the calyceal process around the base of the outer segment. 3D reconstructions (**b**, **c**, **e g–i**), reconstructed coronal slices (**c** and **e** inserts) and optical sections (**d**, **f**, last image of **g–i**, and **j–k**) of cone photoreceptors immunolabeled for the R/G cone opsin (red in **a**, **d–g**; blue in **l**), espin 1 (magenta **e**, **f**), Protocadherin15 (green), cone arrestin (orange in **b–d**), prominin 1 (red in **j**), acetylated alpha tubulin (blue in **g**), centrin 3 (purple in **h**), and VLGR1 (cyan in **i**). **l** normalized intensity across the cellular border. The Pcd15 staining is located in a position external to prominin 1. Numerical data in supplementary data 2. OS outer segment, CP calyceal processes, IS inner segment, NL nuclear layer, S synapse, arrowhead: pcd15 staining expression around the periciliary membrane at the base of the cilia. Scale bars represent 2 µm in (**a**, **c**, **d**) and 1 µm in (**e–j**), and 10 µm in (**b**).

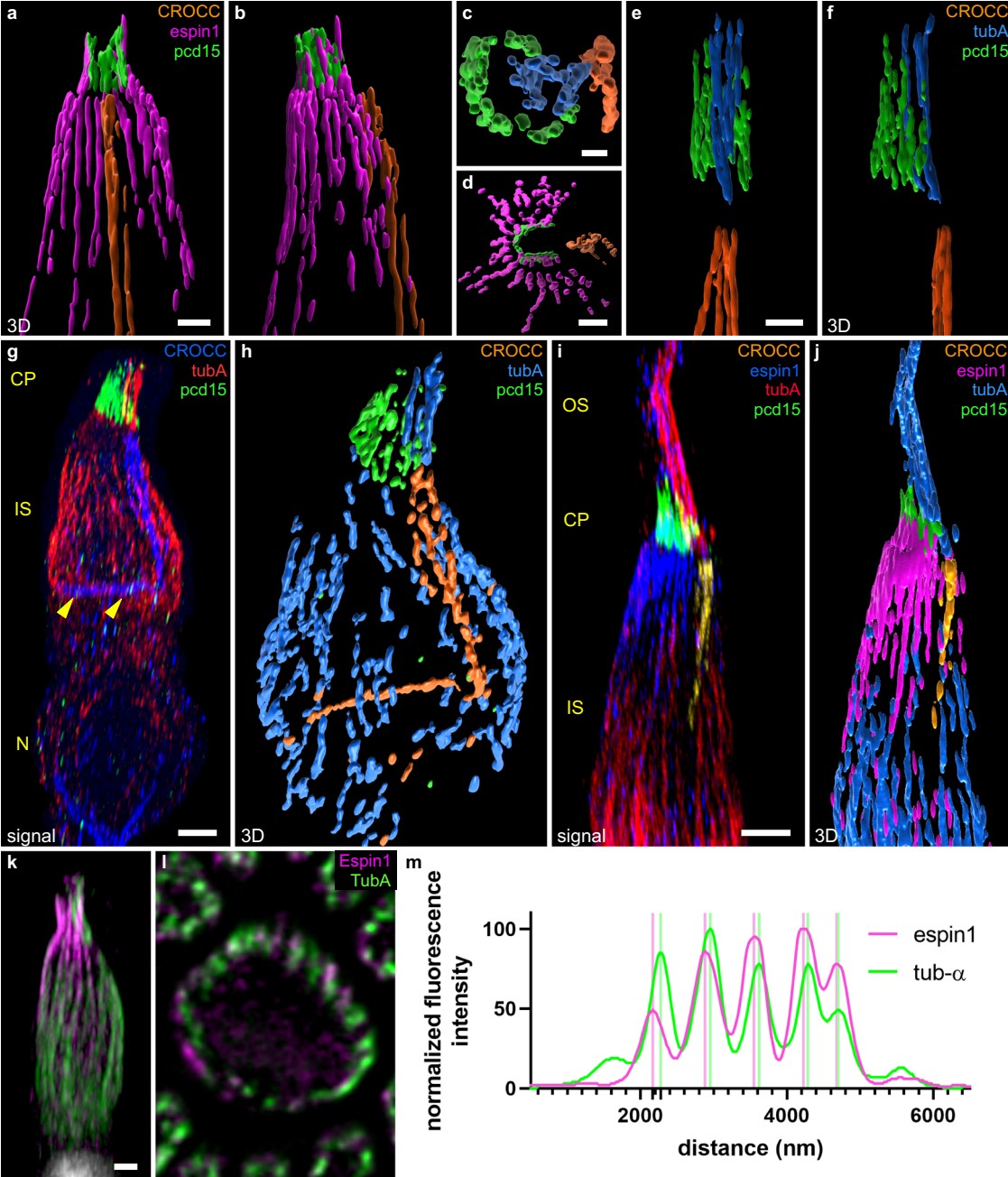

**Fig. 3 Cone cytoskeleton and inner/outer segment alignment.** 3D reconstructions (**a–k**, reconstructed volumes in **a–f**, **h**, and **j**), and an optical slice (**l**) of cones immunolabeled for espin 1 (magenta in all reconstructed volumes: **a**, **b**, **d**, **j**; blue in fluorescence 3D image **l** and magenta in fluorescence 3D image **k**), Protocadherin 15 (pcd15, green), acetylated tubulin alpha (tuba, blue in all reconstructed volumes: **c**, **e**, **f**, **h**, **j**; red in fluorescence 3D images in **g** and **i**; green in **k** and **l**) rootletin (CROCC, orange in all reconstructed volumes: **c**, **e**, **f**, **h**, and **j**, blue in fluorescence 3D image **g** and yellow in fluorescence 3D image **i**). **a**, **b**, **d** and **c**, **e**, **f** represent three perpendicular reconstitutions of the proteins. **g** yellow arrowheads underline the rootletin anchor. An optical slice in the mid-inner segment is shown for a single cone in l, the alternation between espin 1 and tubulin is readily visible both visually and on the fluorescence profile (m, numerical data in supplementary data 3). OS outer segment, CP calyceal processes, IS inner segment, N nucleus. Scale bars represent 0.5 μm in (**c**), 1 μm in (**a**, **d**, **e**), and 2 μm in (**g**, **l**, and **k**).

photoreceptor mosaic displays a similar planar polarity in the primate retina. We attributed an orientation vector to each C-shaped palisade from all the individual cone cells present in a given piece of retina (160 μm × 160 μm) (Fig. 5a). This orientation vector was based on the identification of the C-shaped palisade, by Pcd15 immunolabeling, and of its opening, by immunolabeling of the cilium for acetylated α-tubulin. Surprisingly, an examination of the direction vectors for all such pieces of retina from three different primates showed that the orientation vectors

were always organized along two highly preferred and opposite orientations along a single axis (Fig. 5b). The polar plots clearly showed that the two opposite directions were equally represented, and that the perpendicular orientation was not observed in any of the three samples (Fig. 5c). The C-shaped palisades therefore appeared to be preferentially oriented along a planar axis in these individual pieces of retina. We investigated the definition of these planar axes with respect to location on the retina, by immunolabeling two whole retinas from two different animals and

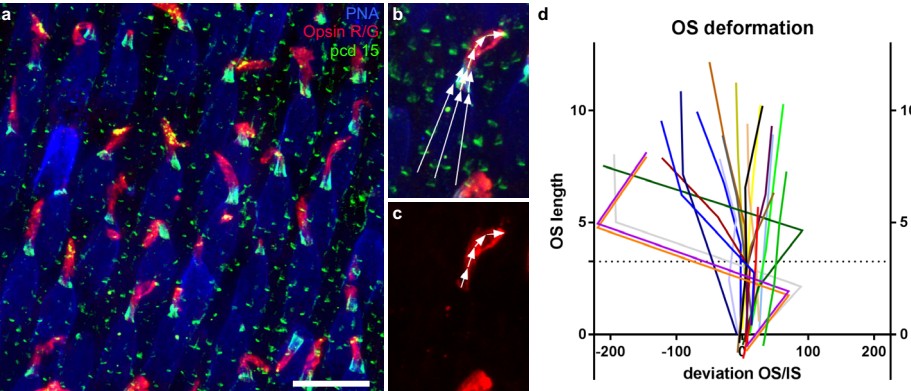

**Fig. 4 Cone inner/outer segment alignment. a–c** Retinal flat-mount immunolabeled for the R/G cone opsin (red), Protocadherin 15 (pcd15, green) and peanut lectin agglutinin (PNA, blue). Definition of the angles between the inner and outer segments on an individual cone (**b, c**). **d** Graph showing the deviation of the angle from linearity for a population of 20 cone outer/inner segments, as a function of the distance to the base of immunostaining for Pcd15 (in degree). The dotted line represented the mean calyceal processes length. (numerical data in supplementary data 4). The scale bar in (**a**) represents 20 μm.

imaging them at regularly spaced locations (Fig. 5d, Sup. fig). The preferred axis was computed, taking into account circular periodicity and the 180° symmetry. This analysis, performed, after masking, by two readers, revealed only minor differences in results between readers, demonstrating the consistency and robustness of the analysis (Fig. 5e). For every coordinate, the calculated axis was reported on the retinal map (Fig. 5f). This map clearly shows the alignment of all but a few planar axes radially, toward a focal central point. The points at which these radii converged were calculated for the two primate retina, and were shown to be either proximal to the fovea, or situated between the fovea and the optic nerve. The very central nature of these positions is consistent with the known projection of the optical axis of the eye onto the retina[35].

**Rod Photoreceptor cytoskeleton.** Calyceal processes have also been described in primate rod photoreceptors[19]. We therefore carefully examined rod photoreceptors by ExM with similar markers, to determine whether this cell type also displayed cytoskeleton architecture compatible with phototropism. Pcd15 staining was distributed in a C-shape at the base of the OS (Fig. 6a–g), beneath which strong rhodopsin or rod transducin immunolabeling of the OS was observed. This conclusion was confirmed by immunostaining of the rod cilium for acetylated α-tubulin (Fig. 6a, b, g) or centrin 3 (Fig. 6c, d). In some cases, the Pcd15 immunostaining extended further into the C-shape (Fig. 6g) and resembled that observed at the base of cone OSs, suggesting expression on the periciliary membrane[36]. This location of Pcd15 opposite the cilium was also observed on immunolabeling for VLGR1, which was present at the opening of the Pcd15-positive C-shape beneath the periciliary membrane (Fig. 6h). Again, as in cones, the number of VLGR1 lobules ranged from 6 to 7. Pcd15 was thus located at the interface between the rod OS and IS. The most surprising finding was the restricted vertical distribution of Pcd15, up to only 500 nm (Fig. 6), whereas it extended up to 3 μm in cone photoreceptors (Fig. 1). Similarly, espin 1 formed no calyceal process-like extensions along the rod OS (Fig. 6i, l; in l, espin 1 appears to be partially colocalized with the tubulin cilium, probably due to an immunohistochemistry artifact, as this pattern was not observed on staining without tubulin antibodies). Espin 1-positive structures appeared to have a limited length within the rod IS. Prominin 1 immunolabeling was too faint for us to successfully visualize by ExM, but the observation of this staining by classic confocal microscopy showed expression to be maximal at the very

base of the rod OS, adjacent to Pcd15 (Fig. 6j, k). Thus, in rods, Pcd15 appeared to be restricted to the point of contact of the disc with the periciliary membrane at the base of the OS. An analysis of the cytoskeleton in rods showed that the filaments were organized differently from those in cones. First, within the IS, espin 1 was limited to the very top of the compartment (Fig. 6i, l). Second, acetylated α-tubulin filaments were present over most of the length of the IS, all around the circumference of the rod (Fig. 6l, m), but they did not seem to be anchored to the base of the IS. Finally, rootletin occupied most part of the IS volume, at its center, surrounded by the acetylated α-tubulin filament, and rooted deeper than the IS, at least to the nuclear compartment (Fig. 6m). Thus, Pcd15 appears to maintain the OS/IS alignment, Hence establishing the optic waveguide property of rod photoreceptors, but the different distribution of cytoskeleton filaments in rods suggests a different functional mechanical role.

## Discussion
Epithelial cells are organized into very well-defined 2D mosaics, following a planar polarity: an organized establishment of asymmetries within cells. This planar polarity is neatly exemplified by the fly retina, with its nine-photoreceptor cell ommatidia, or by the mammalian cochlea, with its four organized rows of hearing cells with aligned V-shaped bundles of stereocilia[37]. In vertebrate, retinal planar polarity has only been described in zebrafish cone photoreceptors, in which all basal bodies and the cilium are decentered on the same polar position[22]. This decentered position is located on the edge of the cell closest to the optic nerve in red-, green-, and blue-sensitive cone photoreceptors. By contrast, no such patterning was observed in ultraviolet-sensitive cones or in rod photoreceptors. This mosaic organization appears to emerge during development, as it is undetectable in larvae seven days after fertilization. We show here that, in the primate retina, the cone photoreceptor cilium also occupies a decentered position at the edge of the cell, either on the side closest to the center of the retina, or on the opposite side. This position on the radius running toward the optic center of the eye, with equal numbers of cilia at each end of the cell radius, may explain why this planar polarity was not described earlier. As for other planar polarities of ciliated cells[37], this specific organization raises considerable questions concerning its establishment, questions lying well beyond the scope of this paper. However, we propose a hypothesis concerning its functional role in relation to the Stiles-Crawford effect, in the orientation of cone photoreceptors toward the optic axis of the eye.

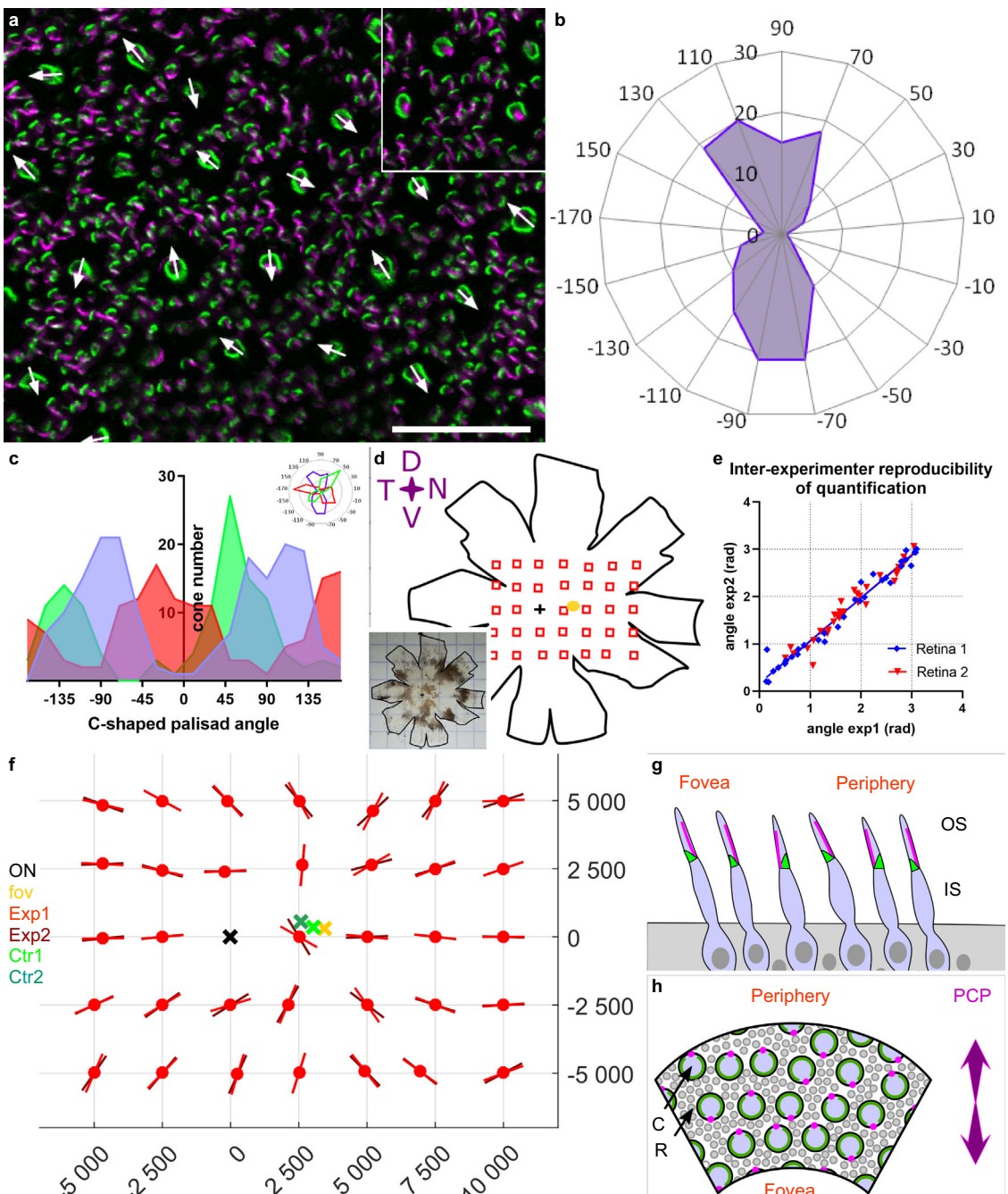

**Fig. 5 Planar polarity of cone photoreceptors in the macaque retina. a** Flat-mounted retina immunostained for protocadherin 15 (pcd15, green) and acetylated tubulin alpha (alpha-Tub, magenta), showing the directions vectors for all the cones in the image. Each cone vector is defined by the C-shaped symmetric axis of the Pcd15 staining and the direction toward the opening of the C-shape defined by the acetylated tubulin alpha staining, as illustrated in the inset. (b) Polar plot of the vectors on a retinal image (0.25 mm²) showing a clear alignment along a preferential planar axis. **c** Graphs representing the distributions of vector directions for 3 independent retinas, showing that all retinal samples had two peaks in opposite directions (numerical data in supplementary data 5). **d**–**f** Cone vector directions for the primate retina are shown in **d**. The planar axes are illustrated on the matrix (f) for all sample points located on the schematic representation of the retina (**d**). For each location, two experimenters independently placed individual cone vectors and calculated the mean planar axis. **e** Mean planar axis angle, demonstrating the reproducibility of the technique. (numerical data in supplementary data 6) **f** Graph providing the angles of the planar axis obtained for macaque 1 and for each observed retinal location in red (different hues for the datasets of different experimenters). Each planar axis was calculated from a minimum of 50 individual cone vector angles. The optic nerve, fovea, and both reconstructed centers (from the two different datasets) are represented by black, yellow, and green crosses, respectively. (macaque n2 in supplementary figure, and numerical data in supplementary data 7). **g**, **h** Schematic representation of the planar polarity organization discovered (**g**: view as a retinal radial section. Cones appear tilted toward the fovea. **h**: view as a flat-mounted retina. C cones, R rods, Green calyceal processes, magenta cilia). See Supplementary figure for comparison of macaque 1 and 2.

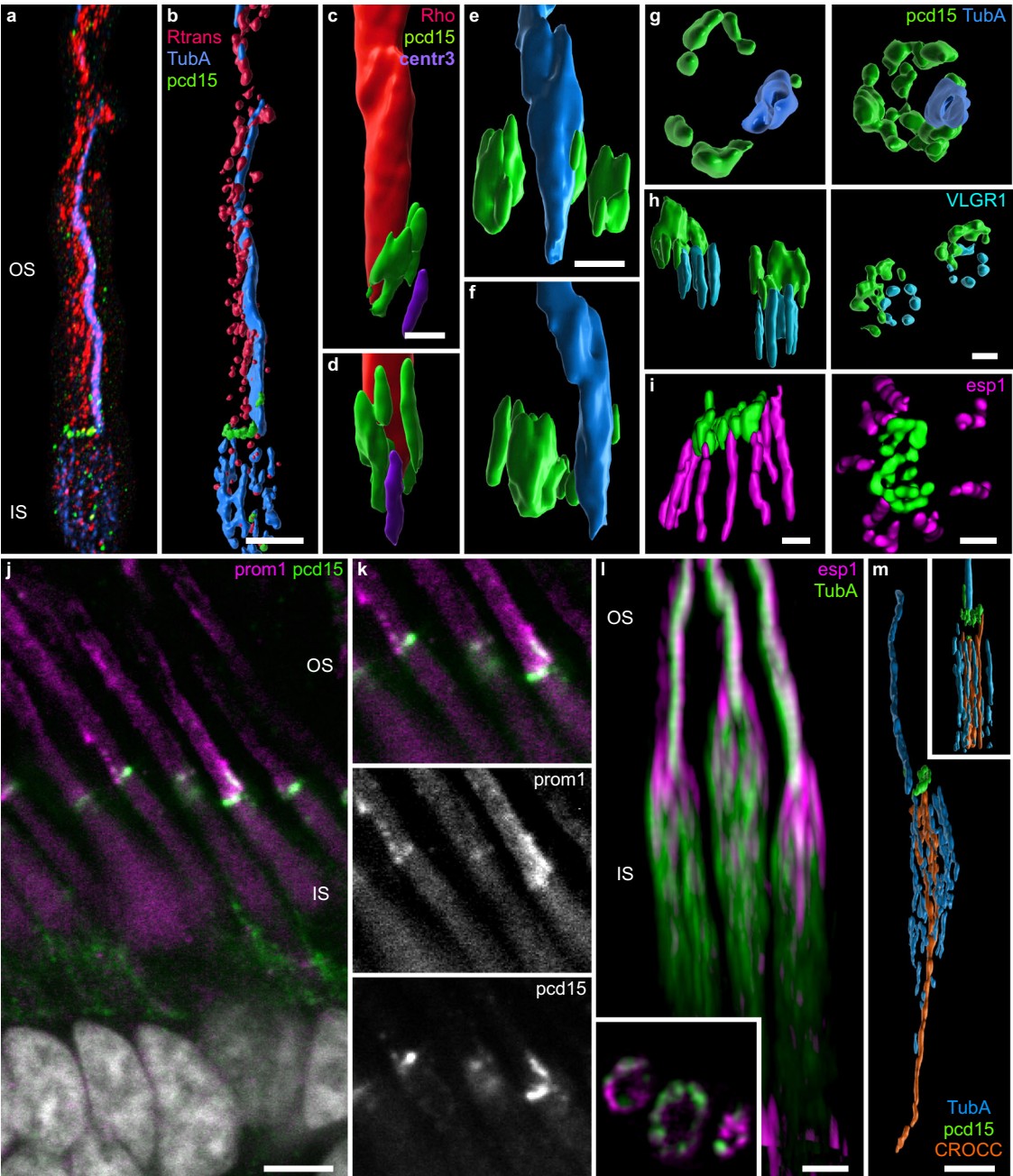

**Fig. 6 Junction of the outer and inner segments of rod photoreceptors.** 3D reconstructions (**a–i**, **l**, **m**) and optical sections (inset **l**) of rod photoreceptors immunolabeled for protocadherin-15 (pcd15, green in all images), acetylated alpha tubulin (TubA, blue in all reconstructed volumes: **b**, **e**, **f**, **g**, **m**, blue in fluorescent 3D image **a**, and green in fluorescent 3D image **l**), centrin 3 (violet in **c**, **d**), rhodopsin (red in **c**, **d**), transducin (red in **a**, **b**), espin 1 (magenta in **l**, **l**), VLGR1 (cyan in **h**), all examined by expansion microscopy (but **l**). **j**, **k** Double-labeling for prominin 1 and pcd15 on retinal slices (noExM). Scale bars after scaling down to take expansion into account are as follows: for **a**, **b**, **l**, **m**: 2 µm, **c**: 1 µm, **d–i**: 0.5 µm); **j**: 5 µm.

The Stiles-Crawford effect implies that photoreceptors have optic wave guide properties, and that those waveguides are correctly aligned in the pupil direction. Using ExM, we show here that Pcd15 is located at the OS/IS interface in both rods and cones. This distribution of Pcd15 is very similar to that previously reported for protocadherin 21 (Pcd21) in mouse rod photoreceptors[38,39]. Ultrastructural studies showed that Pcd21 formed tip-links between the nascent disks and the IS membrane. By analogy to the role of Pcd15 in extracellular membrane links in the hair cells of the cochlea, we suggest that Pcd15, like Pcd21, could play a similar role in the eye, by binding the periciliary membrane and the nascent disks in both rods and cones, to provide strong OS/IS mechanical stability. Such strong

bonds may thus control the growth of the nascent discs at this level revealed by the prominin-1 staining at the OS/IS base. In cones, Pcd15 extends further along with calyceal processes on the OS. No such extension was observed in rods. This morphological difference probably results from structural differences, as cone OSs are composed of membrane microvilli, whereas rod discs are contained in the rod OS membrane. As a consequence, calyceal processes, which were here detected only in cones, may further strengthen the OS/IS interface at the point of nascent microvillus formation[29,40]. A previous study in *Xenopus* showed that Pcd15 (and cd23) knockdown induced both an abnormal bending of the OS, with a rupture of the IS/OS alignment, and irregular lengths for cone OS microvilli[41].

These findings indicate that calyceal processes play two roles: (i) they form a rigid palisade aligning IS and OS, and (ii) they calibrate the size of cone OS discs. Our findings for primate retina are consistent with these results, suggesting that calyceal processes create a tight fixation palisade around the base of the cone OS, where the fragile nascent OS microvilli form, thereby creating a continuous and correctly oriented optic waveguide with the OS/IS by protecting IS to OS alignment.

The different structures of rod OS with disks within the plasma membrane may explain the absence of long calyceal processes in rods. Nascent lamellae were also observed to form from the ciliary membrane at the base of primate rods, but they remained very small, no thicker than 500 nm[42]. This thickness corresponds to the vertical size of Pcd15 immunolabeling in rods. However, we cannot rule out a possible effect of eccentricities, as we worked principally on the peripheral retina, or a fixation artifact, because calyceal processes are highly sensitive to fixation conditions. An absence of calyceal processes has also been reported for mouse photoreceptors, and even for mouse cones[19]. This absence has even been considered to account for the absence of a retinal phenotype in mouse models of Usher syndrome. Another study, based on transmission electron microscopy, reported a single calyceal process in mouse rod photoreceptors[42]. Our results are, therefore, consistent with the notion that rod photoreceptors do not have actin-containing calyceal processes, unless these processes have a molecular composition different from those of cone photoreceptors.

OS/IS continuity is probably reinforced by Usher 2 proteins. In the hair cells of the cochlea, these proteins form ankle links at the base of stereocilia, to strengthen their base. In photoreceptors, these proteins are found at the periciliary membrane or in the extracellular space surrounding the cilium[43]. ExM made it possible to visualize the VLGR1 aggregates in three dimensions, in both rods and cones, around the base of the cilium, confirming the results of more time-consuming immunogold ultrastructural examinations[31]. Our findings are consistent with the notion that VLGR1 establishes a mechanical link between the ciliary membrane and the periciliary membrane, which represents the upper part of the IS and the starting point of the OS. Pcd15 then connects the surface of this periciliary membrane to the first nascent disc or nascent microvillus (see above, illustrated in Fig. 7). These Usher 1 and 2 proteins, thus, tether the IS periciliary membrane to both the cilium and the OS base, maintaining the alignment of the outer and inner segments, despite the expected fragility of this junction (induced by the narrowing of the cell around the cilium). Our analysis of the bending point between the cone IS and OS provides support to this conclusion although these data were gathered on fixed tissues not fully representative of live and in vivo conditions. Usher proteins may therefore compensate for photoreceptor weaknesses at the base of the OS, not only to participate in creating an optic guide, but also to generate a continuous mechanical structure.

The phototropism of photoreceptors required to account for the Stiles Crawford effect implies the presence of a cytoskeleton network capable of adjusting the OS/IS alignment with respect to the optical center of the eye. We provide here the first detailed description of the organization of the cytoskeleton in the primate IS. Our cytoskeleton analysis showed that actin and tubulin formed intertwined cortices within the IS in both types of photoreceptors. These motile filaments were organized around the stable rootletin filaments prolonging the OS cilium within the IS. The stability of rootletin suggests that it establishes a mast within the IS, in the direct continuity with the cilium, thereby creating a rigid structure preventing any deformation other than a slight bending in a given direction. The perpendicular rootletin filament at the IS base may stabilize the rootletin mast further, preventing any displacement, even in response to strong mechanical stress. The position of the cell mast on the radius, toward the optic center of the eye or the macula, provides a greater margin for bending along this axis. The force required to bend the cone OS/IS in a given direction depends on segment length for the projection vector passing through the mast onto the IS base. This force is proportional to the sinus of the angle between the pulling/pushing actin filaments opposite the rootletin mast (see Fig. 7c). In cones, the base of the IS is relatively large with respect to its length, reducing the force required for OS alignment, particularly for large peripheral cones requiring larger displacements. However, rods clearly have a less favorable configuration for producing the same OS tip displacement. For example, a 2 μm shortening of the actin/tubulin cortex in the IS would generate a 0.8 μm displacement for the cone tip (peripheral cone IS diameter: 7.5 μm), whereas the equivalent displacement for a peripheral rod (IS diameter of 2 μm) would be restricted to 0.09 μm. These morphological constraints are consistent with the observation of a Stiles-Crawford effect in cones, but rarely in rods. The absence of a Stiles-Crawford effect in rods must also be considered in light of their functional range in dim light, and, thus, with a dilated pupil. The more central position of the rootletin filament in the rod IS is also consistent with an absence of phototropism in rods.

Previous studies have suggested that the mechanical deformations of photoreceptors are related to the retinomotor movements of photoreceptors during day/night light adaptation in non-mammalian species[44]. Eckmiller previously suggested[45] that the vertical retinomotor movement mechanism reported in fish and amphibian photoreceptors evolved differently in mammals, resulting in a lateral tilting of the photoreceptor OS/IS, mediating the Stiles-Crawford effect. To our knowledge, the photoreceptor rootletin structure is absent in *Xenopus* and teleostean fishes, in which the actin-tubulin cortex lengthens or shortens the inner segment as a whole during retinomotor movements. The asymmetric lateral phototropism expected for the Stiles–Crawford effect implies the involvement of light-activated molecular motors. A transient OS phototropic tilt has been described in photoreceptors[11–15]. However, this initial OS compression towards the direction of the light must be converted into stable morphological changes, and must be transmitted to the underlying compartment, the IS. By analogy to hair cells, we suggest that the Usher1 protein may be involved in mechanotransduction, leading to this stable morphological change. Usher 1 proteins are known to contribute to the mechanotransduction of hearing, with the bending of the stereocilium triggering the opening of calcium channels[46]. In auditory mechanotransduction, Pcd15 forms tip-links or extracellular heterodimers binding two stereocilia with the partner protein cadherin23, such that the stretching of the tip-link results in the opening of the channel[27]. Drawing analogies between photoreceptors and the hair cells of the cochlea, calyceal processes and stereocilia have a similar ontology, suggesting a possible role for calyceal processes in mechanotransduction[30]. The initial transient phototropic tilt of the OS may trigger a mechanical stretching of the Pcd15 links between calyceal processes and the OS membrane. This might lead to a change in the $Ca^{2+}$ channel activity of calyceal processes resulting in either polymerization/depolymerization of the cytoskeleton, or a relative movement of the actin and tubulin cortices (see Fig. 7d, f). This reorganization of the cytoskeleton on the stretched side would generate the morphological change in the IS required for cone OS/IS realignment. Evidence for such mechanotransduction has been obtained for salamander photoreceptors, based on their sensitivity to ultrasound stimulation[47]. Our observations on cytoskeleton organization and especially planar polarity of cone photoreceptors along the eye radius may provide an important step toward the understanding of the preferred phototropism and fine alignment along this axis for the Stiles-Crawford effect. Further studies are now required to

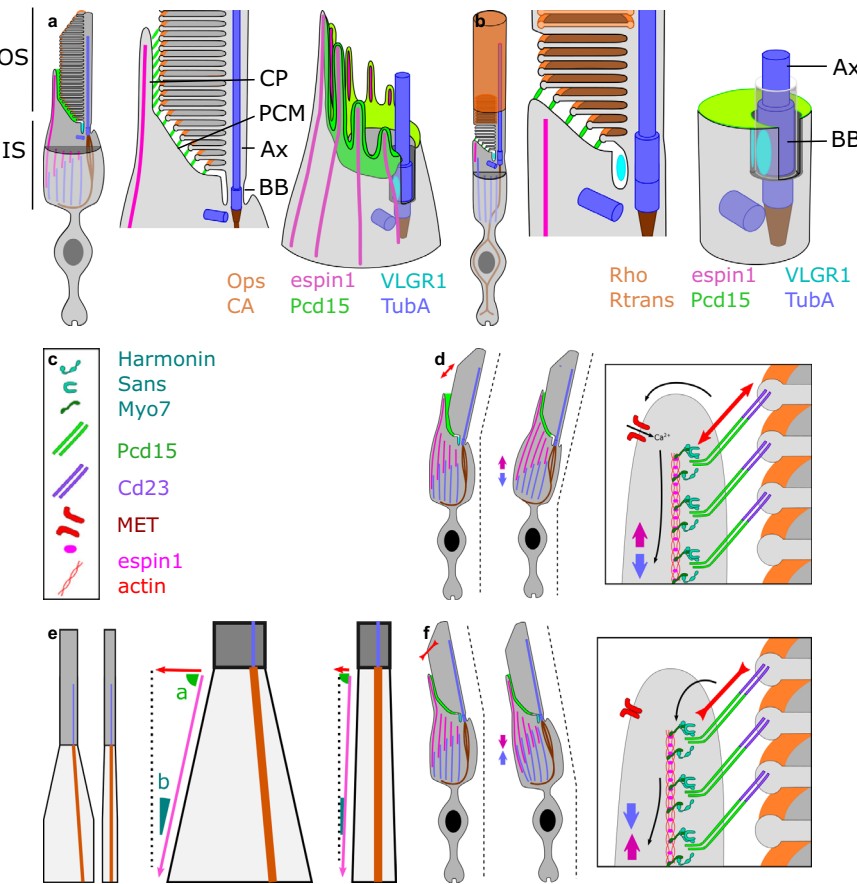

**Fig. 7 Schematic diagram of the distribution of USH proteins, other mechanical proteins, and cilium-associated proteins in rod and cone photoreceptors.** (**a**: cone, **b** rod). Schematic reconstruction of cone 3D inner/outer segment junction structure, showing: full photoreceptor view, a detailed transverse view, and a 3D reconstitution of the apical structure of the inner segment (outer segment remove for visual clarity). **e** Difference between cones and rods in inner-to-outer segment diameter ratio, leading to different effects of identical shortenings of the inner segment. **d**, **f** hypothesis for outer segment tilting leading to a change in inner segment alignment, **c** legend of the symbols are used in panels **d** and **f**. Ax axoneme, BB basal body, CP calyceal process, PCM periciliary membrane, OS outer segment, IS inner segment.

demonstrate the Usher protein involvement to mediate the photoreceptor phototropism at the calyceal processes through a potential mechanotransduction.

## Methods

**Animals**. All the animals used in this study were adult cynomolgus macaques (*Macaca fascicularis*) of foreign origin. The animals were killed ethically and subjected to intracardiac perfusion with 4% paraformaldehyde, and the eyes were then removed.

Ethics approval was obtained from the local ethics committee, CETEA no.44 of MIRCen, and the French Ministry of Education and Research. All experiments on animals were performed by experienced veterinary surgeons, in accordance with national and European laws. The animal facility has authorization from the local authorities to conduct experiments on non-human primates (NHP) (no. D 92 032 02), in accordance with EU directives regarding the use of animals in research (2010/63/EU).

**Retina dissection**. The anterior segment and the vitreous were removed for post-perfusion fixation of the retina by incubation for 1 h at room temperature in 4% PFA (15710, ThermoFisher scientific).

Eyecups were either prepared for cryosection or preserved for retina isolation and whole-mount immunochemistry. For cryosections, retinal samples were cryoprotected (incubation in a series of sucrose solutions with concentrations increasing from 10% to 30% (84097, Sigma-Aldrich)), embedded in OCT medium (6502, ThermoFisher scientific), and sliced on a cryostat. For whole-mount immunochemistry, retinas were separated from other tissues and processed for immunostaining.

**Immunostaining**. Two immunostaining solutions were used, depending on the properties of the antibodies and the reaction: PBS-DTT (donkey serum 5% - Triton 0.5% - Tween 0.5%: Donkey Serum, 017-000-121, Jackson ImmunoResearch; Triton

X100 T8787, Sigma Aldrich; Tween 20, P7949, Sigma Aldrich) or PBSGT (1 x PBS containing 0.2% gelatin: 24350262, Prolabo, and 0.5% Triton X-100), at room temperature, with shaking at 70 rpm[48]. Non-specific binding was blocked and permeabilization was performed in either of these two solutions, at room temperature, for 1 or 2 h. The tissues were then incubated with the primary and secondary antibodies, in the corresponding solution, at 37 °C overnight for the primary antibody and the secondary antibody (see Table 1 for antibodies references).

Samples were then mounted in Vectashield (H1000, Vector Laboratories) for classic confocal microscopy, or processed for expansion microscopy.

**Expansion microscopy protocol**. We performed protein retention expansion microscopy (Pro-ExM) as described by Tillberg et al.[26] and Asano et al.[49]. Briefly, after immunostaining, we anchored the fluorescent markers by overnight incubation with AcX (A20770 – Thermo Fisher Scientific) at room temperature. We then incubated the samples in monomer solution for one hour. Samples were then polymerized at 37 °C for three hours in custom-built chambers, digested with proteinase K and expanded in distilled water.

We modified the initial protocol in three specific ways.

The original protocol required the manipulation of solid acrylamide and N,N-bis acrylamide. As these materials are highly toxic if the powder is inhaled, we adapted the protocol to make use of the liquid form (01697 and M1533, Sigma Aldrich), the toxicity of which is easier to manage.

After polymerization, we used epifluorescence microscopy to determine the orientation of the retina (to identify the side bearing the photoreceptors). We then cut the gel in a non-chiral form ("Pacman" shape). This made it possible to determine unambiguously how the gel should be oriented and to work more easily with deep asymmetric tissues.

Gel movement was problematic for confocal imaging. Gelatin immobilization, as proposed in the initial protocol, did not yield satisfactory results. We found that the gels reacted well with more definitive immobilization. As we were able to determine the optimal orientation without imaging, we did not require a removable immobilization medium. We therefore placed the expanded gels directly on a six-

**Table 1 References of antibodies used in this study and their corresponding dilution.**

**Primary antibody**

| Target | Species | Dilution | Reference | Provider |
|---|---|---|---|---|
| Acetyl-alpha-Tubulin (Lys40) | Rabbit monoclonal | 1/1000 | SAB5600134 | Sigma Aldrich |
| acetyl-alpha tubulin, clone 6-11B-1 | Mouse monoclonal | 1/1000 | MABT868 | Sigma Aldrich |
| Centrin 3 antibody | Rabbit polyclonal | 1/500 | ab228690 | Abcam |
| Opsin Antibody, Red/Green | Rabbit polyclonal | 1/200 | AB5405 | Sigma Aldrich |
| Espin 1 | Mouse monoclonal | 1/200 | sc-515657 | Santa Cruz Biotechnology |
| human Cone arrestin | Rabbit polyclonal | 1/2000 | LUMIf-hCAR/human cone arrestin (ARR4), C. M. Craft, University of Southern California Roski Eye Institute, Los Angeles, CA | |
| Protocadherin-15 | Sheep polyclonal | 1/500 | AF6729 | Biotechne |
| Prominin 1 | Mouse monoclonal | 1/200 | MA1219 | ThermoFisher |
| Rhodopsin, clone RET-P1 | Mouse monoclonal | 1/400 | MAB5316 | Sigma Aldrich |
| Gα t1, Rod transducin | Mouse monoclonal | 1/1000 | sc-136143 | Santa Cruz Biotechnology |
| CROCC (Rootletin) | Rabbit polyclonal | 1/400 | NBP180820 | Biotechne |
| VLGR1 | Rabbit polyclonal | 1/400 | C terminal extremity, aa 6149–6298, accession no. Q8VHN7; V2CD antibody, A. El-Amraoui and C. Petit, Institut Pasteur, Paris, France. | |

**Secondary antibody**

| Target | Species | Alexa | Reference | Provider |
|---|---|---|---|---|
| Mouse | Donkey polyclonal | 488 | A-21202 | Invitrogen |
| Rabbit | Donkey polyclonal | 488 | A-21206 | Invitrogen |
| Sheep | Donkey polyclonal | 488 | A-11015 | Invitrogen |
| Mouse | Donkey polyclonal | 594 | A-21203 | Invitrogen |
| Rabbit | Donkey polyclonal | 594 | A-21207 | Invitrogen |
| Sheep | Donkey polyclonal | 594 | A-11016 | Invitrogen |
| Mouse | Goat polyclonal | 633 | A-21052 | Invitrogen |
| Rabbit | Goat polyclonal | 633 | A-21070 | Invitrogen |
| Sheep | Donkey polyclonal | 633 | A-21100 | Invitrogen |
| Mouse | Donkey polyclonal | 647 | A-31571 | Invitrogen |
| Rabbit | Donkey polyclonal | 647 | A-31573 | Invitrogen |

All secondary antibodies where used at a 1/500 dilution for noExM and 1/250 dilution for ExM staining protocols.

well glass-bottomed plate (P06-20-1.5-N. 220.200.022 IBL Cellvis), to which we added Sylgarde (H05217B145, Dow Corning) on top and around the gel, to encapsulate the gel completely in an airtight matrix. This strategy made it possible to eliminate gel movement completely during image acquisition.

The ExM experiment was performed in parallel with classic confocal immunochemistry (noExM) on consecutive retina pieces, to check for ExM-induced artifacts and immunochemistry problems.

**Confocal imaging**. All images were acquired with a confocal microscope, with either an oil immersion objective (classical imaging) or a silicon objective (allowing deeper imaging, expansion microscopy).

**Image processing**. Images, 3D volumes were generated with FIJI and Imaris x64 software (version 8.4.1 and 9.2.1 Bitplane). Stack images were first converted to imaris files (.ims) with ImarisFileConverter, and 3D reconstruction was performed with the 'volume rendering' function. Optical slices were obtained with the 'orthoslicer' tool. The surface tool was applied manually applied, with selection of the mask option, to isolate individual cells (i.e., isolated cone photoreceptor). 3D pictures and movies were generated with the 'snapshot' and 'animation' tools.

For fluorescence profile, we used the 'plot profile' tool on FIJI, and normalized intensity on the obtained maximum for each channel.

**Statistics and reproducibility**. For planar angle calculation within the retinal map, the preferred axis was computed considering circular periodicity and 180° symmetry. For our calculation, we used a bootstrap with 20 thousand samples to estimate the standard error of the mean. 34 locations were chosen on each retina following a regular grid pattern (Fig. 5f). For each location, two experimenters independently placed individual cone vectors (while blind for the location of the image) to calculated the mean planar axis, with a minimum of 50 individual cones in one image. Two retinas were used to confirm results from two different macaques.

GraphPad prism and Matlab were used for statistical analyses.

**Reporting summary**. Further information on research design is available in the Nature Research Reporting Summary linked to this article.

## Data availability

The data that support the main findings of this study are openly available in figShare: https://figshare.com/projects/Planar_polarity_in_primate_cone_photoreceptors_a_role_in_Stiles_Crawford_effect_phototropism_/124927.

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

## Acknowledgements

We thank the imaging platforms at the *Institut de la Vision*. We thank Cheryl Mae Craft (University of Southern California ROSKI Eye Institute, Los Angeles, CA) for providing the antibody (LUMIf-hCAR) against cone arrestin, Christine Petit and Aziz El-Amraoui for their insight into Usher syndrome and for providing the anti-VLGR1 antibody. We thank Michaël Trichet from the IBPS electron microscopy core facility and the «Région Île-de-France», Sorbonne-Université and the CNRS for support. We thank the primate platform of the Molecular Imaging Research Center (MIRCen) platform at Fontenay-aux-Roses: Claire-Maëlle Fovet, Joanna Demilly and Francis Relouzat; and Hanen Khabou and Deniz Dalkara who provided us with some of the primate tissues used. This work was supported by the European Research Council (ERC) Synergy Grant Scheme (holistic evaluation of light and multiwave applications to high-resolution imaging in ophthalmic translational research revisiting the Helmholtzian synergies, HELMHOLTZ ERC Grant Agreement 610110), French state funds managed by the *Agence Nationale de la Recherche* within the *Investissements d'Avenir* program, RHU LIGHT4DEAF [ANR-15-RHU-0001], LABEX LIFESENSES [ANR-10-LABX-65], and IHU FOReSIGHT [ANR-18-IAHU-0001].

## Author contributions

A.V., S.P. and J.A.S. conceived and initiated the project. A.V. and S.P. designed research; A.V. and L.B. performed research; U.F., S.F., and C.N.J. contributed new reagents/analytic tools; A.V. analyzed data; A.V., S.P. and J.A.S. wrote the paper.

## Competing interests

The authors declare no competing interests.

## Additional information

**Peer Review Information** *Communications Biology* thanks the anonymous reviewers for their contribution to the peer review of this work. Primary Handling Editors: Simona Chera and Karli Montague-Cardoso. Peer reviewer reports are available.

