## [Peer Review File · Communications Biology]

Reviewers' comments:

Reviewer #1 (Remarks to the Author):

The manuscript by Verschuere et al., describes a possible mechanism for the Stiles Crawford effect. The mechanism is based on high resolution expansion microscopy examining the periciliary margin and calyceal processes of cones and rods in macaque retina. The primary result based on staining of tubulin, actin, protocadherins, together with outer segment proteins and other structural proteins located at the IS/OS boundary, is a beautifully detailed high-resolution molecular organization of this critical cellular region in rods and cones. The authors then hypothesize, based on vectorial measurements of the IS/OS boundary and the conservation of function of proteins, a mechanism for tilting of cones toward the optical center of the eye (phototropism).

I thought that this manuscript was very interesting with high quality images and I only noted some typos, omissions that should be corrected.

1. line 46. Not clear if sentence isn't finished or if the word "as" needs to be deleted.
2. line 72. "has been" is repeated twice.
3. line 152. I thought the word "stained" should be inserted between "intensely" and "cone".
4. line 153. Is the word "prolongs" correct here? Would "provides" be better?
5. Figure 2. CP and PC should be defined in the legend. Are these the same or different?
6. Figure 3. For "a" through "f" it would be helpful to indicate label with color code in figure panels (similar to h, j and l).
7. Figure 4. Color coding in figure panels would be helpful to reader.
8. Figure 4. Dotted line should be defined in the figure legend.
9. Figure 5. Line 633 refers to primate retina shown in "E", but there isn't a panel "E".
10. Figure 5. Panels "g" and "h" are not described in the legend.
11. Figure 5. Panel "a" should contain color-coded labels.
12. Figure 7. Line 656 and 657 list many abbreviations which are omitted from the figure.

Reviewer #2 (Remarks to the Author):

The Stiles-Crawford effect, caused by the alignment of photoreceptor inner and outer segment orienting toward the pupil, is known to be dynamically phototropic. The mechanism responsible for this movement is not well understood. The manuscript by Verschuere undertook a high-resolution microscopy study using expansion microscopy focusing on Usher proteins and other cytoskeletal elements to address this question. The images presented are very beautiful, and the finding that the primate cone photoreceptors have a planar polarity organized radially around the optical center of the eye is interesting. However, the study is descriptive in nature and much of the information on the cytoskeletal and Usher proteins have been previously reported. These issues limit the insight provided on the mechanism behind the Stiles-Crawford effect.

1. The manuscript contains numerous language errors (e.g. line 46, line 72).
2. The labeling of images is inconsistent. For example, calyceal processes (CP) is sometimes mislabeled as PC in figure 2 and 3.
3. Labeling of figure 2 panels do not fit with the description of the figure legend or the manuscript text.
4. Often the abbreviations in figures are not explained.
5. The switching of the coloring for the same antibody staining in different panels of the figure is confusing. What are the orange arrowheads pointing at in Fig. 3g?
6. The testing of the hypothesis that OS/IS junction formed a very rigid and structure using fixed retina sample (Fig. 4) is problematic.

Reviewer #3 (Remarks to the Author):

The manuscript by Verschuere and colleagues provides a characterization of the structure of the ciliary/calyceal regions of primate photoreceptors at very high level of resolution. Using a combination of immunohistochemistry for confocal microscopy and expansion microscopy the authors describe the structure of the inner segment-outer segment boundary in both cones and rods. Importantly, they show that the key proteins associated with the cytoskeleton and ciliary specialization are aligned in a way to hold the inner segment-outer segment boundary fixed and rigid in cones. Furthermore, the alignment of these structures was maintained when the retina was deformed, even while outer segment could be deformed. In whole mount retina the authors also show that the polarity of calyceal processes were always oriented toward two retinal locations, showing a directional organization reminiscent of cochlear hair cells. Finally, the authors describe the structural differences with the ciliary region of rod photoreceptors, and discuss these differences. Overall, this work reinforces the role of the Usher protein, protocadherin15, as a critical part of the cilium structure.

It should be noted that the findings of this manuscript on their own are significant and well done. This work describes the ciliary region of primate cones at an unprecedented spatial resolution, compares these with rods, and will be important for future work studying the formation and maintenance of outer segments in health and disease. That being said, the weakness in the manuscript is largely from the broader speculations relating to potential phototropisms and the relative role of these calyceal structures in the Stiles-Crawford effect. Waveguide properties of cone photoreceptors are complex and are known to also require their pedicle structure and the lensing effects of mitochondria in the ellipsoid region. The rigid structure of the inner segment-outer segment boundary may simply be the next stage of alignment of the cone to the optical axis. Similarly, the proposed mechanism for motor movements is not supported by any physical measurements or manipulations, and as such it is not clear whether it can explain the magnitude of the movement required to permit the extent of phototropism to explain the Stiles-Crawford Effect. The overly speculative nature of the manuscript in these two areas diminishes enthusiasm for an otherwise technical tour de force.

Major comments:

1. While the data presented in figure 4 provides some insight of the mechanical link at the transition between outer and inner segments, it is very difficult to interpret these data as they lack proper calibration of the amount of force provided in the manipulation.
2. The general narrative of the paper is perhaps misleading. The authors provide no direct evidence that the described structures and proteins are in fact responsible for the Stiles-Crawford effect.
3. The authors interpretation of the role of the investigated calyceal structures relating to phototropisms and structural support is not well established. To what extent do these structures provide rigidity while enabling the necessary movement?

Minor comments:

1. The final sentence of the Introduction first paragraph is missing its second half.
2. Some Figure labels are missing, such as for 2l, 3m. Please make sure Figure Legends describe all the panels.
3. The title of the manuscript overstates the significance of the findings relating to the Stiles-Crawford effect (see text above).

Answer to the referee's review.

All changes were highlighted in yellow in the text.

Referee expertise:

Referee #1: biology of the cone photoreceptor, molecular basis of human retinal disease

Referee #2: Retina, Rod and Cone Photoreceptors, Vision, G protein signal transduction

Referee #3: visual sensitivity, retinal circuitry, Rod and Cone Photoreceptors

Reviewers' comments:

Reviewer #1 (Remarks to the Author):

The manuscript by Verschueren et al., describes a possible mechanism for the Stiles Crawford effect. The mechanism is based on high resolution expansion microscopy examining the periciliary margin and calyceal processes of cones and rods in macaque retina. The primary result based on staining of tubulin, actin, protocadherins, together with outer segment proteins and other structural proteins located at the IS/OS boundary, is a beautifully detailed high-resolution molecular organization of this critical cellular region in rods and cones. The authors then hypothesize, based on vectorial measurements of the IS/OS boundary and the conservation of function of proteins, a mechanism for tilting of cones toward the optical center of the eye (phototropism).

I thought that this manuscript was very interesting with high quality images and I only noted some typos, omissions that should be corrected.

review	answer
1. line 46. Not clear if sentence isn't finished or if the word "as" needs to be deleted.	'As' needed to be deleted, we corrected this. Line 47
2. line 72. "has been" is repeated twice.	We corrected this. Line 73
3. line 152. I thought the word "stained" should be inserted between "intensely" and "cone".	Indeed, we corrected this. Line 153
4. line 153. Is the word "prolongs" correct here? Would "provides" be better?	Not exactly, we propose that USHER2 proteins would establish the link between OS and IS at the very base of the cilia, and USHER1 would establish the link between OS and IS around the periciliary membrane and nascent cone disks. We changed the sentence in the text for more clarity. (line 154 - 158)
5. Figure 2. CP and PC should be defined in the legend. Are these the same or different?	They are the same, and have been corrected to CP in all figures.
6. Figure 3. For "a" through "f" it would be helpful to indicate label with color code in figure panels (similar to h, j and l).	We added color coding in figure 3 a to j panels.
7. Figure 4. Color coding in figure panels would be helpful to reader.	We added color coding in figure 4.
8. Figure 4. Dotted line should be defined in the figure legend.	We added 'The dotted line represented the mean calyceal processes extension' to the figure legend. l648
9. Figure 5. Line 633 refers to primate retina shown in "E", but there isn't a panel "E".	'e' was switched to 'd' l658
10. Figure 5. Panels "g" and "h" are not described in the legend.	We added: "(g, h) : schematic representation of the planar polarity organization discovered (g : view as a retinal radial section. Cones appear tilted toward the fovea. h : view as a flat-mounted retina. C : cones, R : rods. Green : calyceal processes, magenta : cilia)" l664-665
11. Figure 5. Panel "a" should contain color-coded labels.	We added color coding in panel A.
12. Figure 7. Line 656 and 657 list many abbreviations which are omitted from the figure.	We added those abbreviations to the figure7.

Reviewer #2 (Remarks to the Author):

The Stiles-Crawford effect, caused by the alignment of photoreceptor inner and outer segment orienting toward the pupil, is known to be dynamically phototropic. The mechanism responsible for this movement is not well understood. The manuscript by Verschueren undertook a high-resolution microscopy study using expansion microscopy focusing on Usher proteins and other cytoskeletal elements to address this question. The images presented are very beautiful, and the finding that the primate cone photoreceptors have a planar polarity organized radially around the optical center of the eye is interesting. However, the study is descriptive in nature and much of the information on the cytoskeletal and Usher proteins have been previously reported. These issues limit the insight provided on the mechanism behind the Stiles-Crawford effect.

review	answer
1. The manuscript contains numerous language errors (e.g. line 46, line 72).	All reported and found language errors were corrected (and especially l47 and 73).
2. The labeling of images is inconsistent. For example, calyceal processes (CP) is sometimes mislabeled as PC in figure 2 and 3.	The labeling was specifically checked. CP is now labelled consistently within the document. Changes :  - figure 2: all 'PC' mislabeled were switched to 'CP' - figure 3: all 'PC' mislabeled were switched to 'CP'
3. Labeling of figure 2 panels do not fit with the description of the figure legend or the manuscript text.	We checked both the figure legend and manuscript text and corrected the labeling.
4. Often the abbreviations in figures are not explained.	We checked all legends to add the missing figures' abbreviations explanation (example l 627-628 and l685-686).
5. The switching of the coloring for the same antibody staining in different panels of the figure is confusing. What are the orange arrowheads pointing at in Fig. 3g?	We choose to represent the initial fluorescent images (g,h) using only the three conventional colors (and additionally adding yellow for panel Fig.3l where four staining were observed). When representing calculated volumes (Fig. 3h and 3j) we used the same color code as in all figures and importantly for the schematic drawings in Fig. 7. We added the following sentence in the legend of Fig 3g: '(g) yellow arrowheads underline the rootletin anchor.' (l638)
6. The testing of the hypothesis that OS/IS junction formed a very rigid and structure using fixed retina sample (Fig. 4) is problematic.	To comply with the Reviewer's comment, we added in the text that "this observation on fixed tissues cannot represent the in situ live condition". line 194-196 and in the discussion (l323-324).

Reviewer #3 (Remarks to the Author):

The manuscript by Verschueren and colleagues provides a characterization of the structure of the ciliary/calyceal regions of primate photoreceptors at very high level of resolution. Using a combination of immunohistochemistry for confocal microscopy and expansion microscopy the authors describe the structure of the inner segment-outer segment boundary in both cones and rods. Importantly, they show that the key proteins associated with the cytoskeleton and ciliary specialization are aligned in a way to hold the inner segment-outer segment boundary fixed and rigid in cones. Furthermore, the alignment of these structures was maintained when the retina was deformed, even while outer segment could be deformed. In whole mount retina the authors also show that the polarity of calyceal processes were always oriented toward two retinal locations, showing a directional organization reminiscent of cochlear hair cells. Finally, the authors describe the structural differences with the ciliary region of rod photoreceptors, and discuss these differences. Overall, this work reinforces the role of the Usher protein, protocadherin15, as a critical part of the cilium structure.

It should be noted that the findings of this manuscript on their own are significant and well done. This work describes the ciliary region of primate cones at an unprecedented spatial resolution, compares these with rods, and will be important for future work studying the formation and maintenance of outer segments in health and disease. That being said, the weakness in the manuscript is largely from the broader speculations relating to potential phototropisms and the relative role of these calyceal structures in the Stiles-Crawford effect. Waveguide properties of cone photoreceptors are complex and are known to also require their pedicle structure and the lensing effects of mitochondria in the ellipsoid region. The rigid structure of the inner segment-outer segment boundary may simply be the next stage of alignment of the cone to the optical axis. Similarly, the proposed mechanism for motor movements is not supported by any physical measurements or manipulations, and as such it is not clear whether it can explain the magnitude of the movement required to permit the extent of phototropism to explain the Stiles-Crawford Effect. The overly speculative nature of the manuscript in these two areas diminishes enthusiasm for an otherwise technical tour de force.

Answer: To follow the reviewer's concern, we downplayed the conclusion presenting our hypothesis as a mere suggestion. Furthermore, we introduced a reference on the role of mitochondria as a lensing effect in the ellipsoid (line 42-44).

Major comments:

review	answer
1. While the data presented in figure 4 provides some insight of the mechanical link at the transition between outer and inner segments, it is very difficult to interpret these data as they lack proper calibration of the amount of force provided in the manipulation.	To comply with the Reviewer's comment, we underlined the uncontrolled nature of the shear stress applied (line 187-188).
2. The general narrative of the paper is perhaps	We made more explicit that we propose a

misleading. The authors provide no direct evidence that the described structures and proteins are in fact responsible for the Stiles-Crawford effect.	“comprehensive hypothesis” for the Stiles-Crawford phototropism, and downplayed the narrative in the summary (l26 - 48) and the conclusion (l382-386). We added in the introduction that other cellular mechanisms (e.g; mitochondria) are known to be responsible for the waveguide properties of PR (l43-44).
3. The author interpretation of the role of the investigated calyceal structures relating to phototropisms and structural support is not well established. To what extent do these structures provide rigidity while enabling the necessary movement?	This is indeed outside of the current article scope, which is to propose a mechanism for futures studies to build on, but would be the next logical step. We checked the narrative to make this more explicit (l324-326, l382-383).

Minor comments:

review	answer
1. The final sentence of the Introduction first paragraph is missing its second half.	“As” needed to be deleted, we corrected this (l47).
2. Some Figure labels are missing, such as for 2l, 3m. Please make sure Figure Legends describe all the panels.	We checked all figures labels, 2l and 3m were added.
3. The title of the manuscript overstates the significance of the findings relating to the Stiles-Crawford effect (see text above).	We propose a new title as an alternative: “Planar polarity in primate cone photoreceptors: a role in Stiles Crawford effect phototropism?” to highlight that the main findings of the paper are the planar polarity and cytoskeleton architecture but that it provides a hypothesis for a role in Stiles Crawford effect phototropism.

REVIEWERS' COMMENTS:

Reviewer #2 (Remarks to the Author):

The authors addressed my concerns. No more comments.

Reviewer #3 (Remarks to the Author):

The manuscript by Verschueren and colleagues has been reoriented in a manner that highlights the important findings of the work without over-speculation about the relationship between the ciliary/calyceal structure and the Stiles-Crawford effect. In our previous review we emphasized the important technical advance of this work, and the high resolution images of this cellular region. Our main concern was not the data presented, but rather the spin toward functional implications which could not be supported fully. We are pleased with the modifications presented by the authors, including with the manuscript title, and only have some small residual comments.

Minor comments:

1. Line 39 – missing 'to' between thought and be
2. Line 151 – intensely-stained is a compound adjective and should be hyphenated.
3. Line 246 – it seems the word 'part' should have been deleted.
4. Line 296 – correctly-oriented optic waveguide, the first two words in the compound adjective should be hyphenated.

Response to reviewers :

Reviewer #2 (Remarks to the Author):

The authors addressed my concerns. No more comments.

Reviewer #3 (Remarks to the Author):

The manuscript by Verschueren and colleagues has been reoriented in a manner that highlights the important findings of the work without over-speculation about the relationship between the ciliary/calyceal structure and the Stiles-Crawford effect. In our previous review we emphasized the important technical advance of this work, and the high resolution images of this cellular region. Our main concern was not the data presented, but rather the spin toward functional implications which could not be supported fully. We are pleased with the modifications presented by the authors, including with the manuscript title, and only have some small residual comments.

Minor comments:

1. Line 39 – missing ‘to’ between thought and be	‘to’ added, see line 41
2. Line 151 – intensely-stained is a compound adjective and should be hyphenated.	Taken into account, see line 154
3. Line 246 – it seems the word ‘part’ should have been deleted.	Taken into account, line 248
4. Line 296 – correctly-oriented optic waveguide, the first two words in the compound adjective should be hyphenated.	Taken into account, see line 292